# Anion Exchange Membrane Based on BPPO/PECH with Net Structure for Acid Recovery via Diffusion Dialysis

**DOI:** 10.3390/ijms24108596

**Published:** 2023-05-11

**Authors:** Haiyang Shen, Yifei Gong, Wei Chen, Xianbiao Wei, Ping Li, Congliang Cheng

**Affiliations:** 1School of Materials & Chemical Engineering, Anhui Jianzhu University, Hefei 230022, China; 2Department of Mathematics & Physics, Anhui Jianzhu University, Hefei 230022, China

**Keywords:** BPPO, PECH, anion exchange membrane, diffusion dialysis, acid recovery, net structure

## Abstract

In order to improve the performance of the anion exchange membrane (AEM) used in acid recovery from industrial wastewater, this study adopted a new strategy in which brominated poly (2,6-dimethyl-1,4-phenyleneoxide) (BPPO) and polyepichlorohydrin (PECH) were used as the polymer backbone of the prepared membrane. The new anion exchange membrane with a net structure was formed by quaternizing BPPO/PECH with N,N,N,N-tetramethyl-1,6-hexanediamine (TMHD). The application performance and physicochemical property of the membrane were adjusted by changing the content of PECH. The experimental study found that the prepared anion exchange membrane had good mechanical performance, thermostability, acid resistance and an appropriate water absorption and expansion ratio. The acid dialysis coefficient (U_H_^+^) of anion exchange membranes with different contents of PECH and BPPO was 0.0173–0.0262 m/h at 25 °C. The separation factors (S) of the anion exchange membranes were 24.6 to 27.0 at 25 °C. Compared with the commercial BPPO membrane (DF-120B), the prepared membrane had higher values of U_H_^+^ and S in this paper. In conclusion, this work indicated that the prepared BPPO/PECH anion exchange membrane had the potential for acid recovery using the DD method.

## 1. Introduction

With the development of industry, an amount of industrial waste is produced, which brings environmental pollution, climatic anomalies and other problems [1]. Therefore, the recycling of industrial waste contributes to the sustainable development of the economy, the protection of the ecological environment, and the efficient utilization of resources, which is of great significance for the improvement of human life. Acid wastewater mainly comes from industrial production processes such as electroplating [2], metallurgy and mining [3]. Acid wastewater has important recycling and utilization value because of its large quantity, high industrial utilization value and easy recovery. How to treat acid wastewater generated in industrial production process is still the main concern of researchers [4,5].

The latest method of treating industrial waste acid is membrane separation technology [5]. As the core component of membrane separation technology, the ion exchange membrane (IEM) has attracted extensive attention in scientific fields due to its selective permeability to specific ions [6,7,8]. IEM is mainly divided into two categories: anion exchange membrane (AEM) and cation exchange membrane (CEM). AEM includes positively charged fixed groups and CEM includes negatively charged fixed groups [7,8,9]. AEM and CEM are often used in electrodialysis (ED) [10], pervaporation (PV) [11] and diffusion dialysis (DD) [12]. Compared with other membrane separation processes, the DD process has the advantages of low pollution, convenient equipment installation and low equipment operation costs [13,14]. The working principle of DD is driven by the concentration difference, which transfers ions from the high concentration side of the membrane to the low concentration side [14]. As the core component of DD, AEM has high ion penetration selectivity, reasonable expansion ratio and high mechanical stability.

The main influence factor of the DD process is the performance of AEM. Therefore, the preparation of an AEM with excellent performance (such as good acid resistance, heat stability, mechanical properties, ion transport performance, etc.) would help further improve the acid recovery efficiency of the DD process and reduce energy consumption [7]. Many kinds of materials are used to manufacture anion exchange membranes, such as polysulfone (PSF) [15], polyvinyl alcohol (PVA) [16], chitosan (CS) [17], polybenzimidazole (PBI) [18], brominated poly (2,6-dimethyl-1,4-phenyleneoxide) (BPPO) [19] and other materials [20,21,22]. The anion exchange membrane consists of appropriate polymer materials, which can achieve good separation effects in acid recovery using DD. Among the different polymer materials, poly (2,6-dimethyl-1,4-phenyleneoxide) (PPO) is widely applied in preparing AEM because of its good properties and the easy preparation of the membranes after bromination treatment [23,24,25]. The main problem associated with AEM is its hydrophily, which limits the ion transport rate in the DD process [26,27]. Currently, PECH is a common polymer material that has been little researched in terms of anion exchange membranes [28,29,30]. The reason is that the highly quaternized PECH membrane expands severely in water, which influences the application of PECH in DD [30]. Due to the low hydrophily of BPPO membrane, we plan to crosslink PECH and BPPO with TMHD to improve the membrane performance. In this work, PECH and BPPO were combined to overcome their respective disadvantages. The quaternization of PECH enhances the hydrophilicity of the membrane and its ion permeability. The quaternization BPPO improves the mechanical property of the membrane [31,32]. At the same time, PECH and BPPO form a net structure by quaternization, which enhances the heat resistance of the membrane [31,32,33].

The AEM for the DD process was prepared using the quaternization reaction of TMHD with BPPO and PECH. The performance of the membrane was improved through the quaternization reaction of TMHD with BPPO and PECH. By changing the content of PECH in the membrane, a series of membranes with different quaternization degrees was prepared. The infrared spectral, ion exchange rate, water absorption, TGA and morphology of the membranes were presented. The application effect of the prepared membranes was also tested in acid recovery via the DD process. The results showed that the prepared membranes have potential in acid recovery using DD.

## 2. Results and Discussion

### 2.1. FTIR Spectra of the Synthesized Membranes

In order to verify the successful preparation of the BPPO/PECH ion exchange membranes, the membranes were tested by FTIR. The FTIR spectra data from the BPPO/PECH film and membranes 1–5 were collected and are shown in Figure 1. According to Figure 1, the infrared absorption peak at 3200–3500 cm^−1^ was the infrared peak of water. The peak of water was caused by the residual water in the membrane, which showed the hydrophilicity of the membrane [25]. The peak at 2920–2860 cm^−1^ was the infrared absorption peak of methyl [34,35,36]. The infrared peak at 1025–1303 cm^−1^ was the infrared absorption peak of the symmetrical and asymmetrical stretching vibrations of C-O in BPPO [34,35]. The characteristic peak of the C-N bonds in the membrane was at 1610 cm^−1^ [36]. The reduction of the FTIR peak at 700 cm^−1^ and the presence of a characteristic peak at 1610 cm^−1^ represent the successful quaternization reaction of PECH and BPPO with THMD [35,36].

### 2.2. SEM Analysis of Membrane Morphology

The changes in the cross-section morphologies of membranes 1–5 were observed by SEM. The SEM images of membranes 1–5 were collected and are presented in Figure 2. The microstructure of the membrane was one of the important factors affecting the performance of the membrane. From Figure 2, it can be seen that membranes 1–5 experienced slight phase separation, which allowed more ions to pass through membrane; thus, the efficiency of the acid recovery was enhanced [36,37]. At the same time, the Fe^2+^ ion transportation rate was also increased [36]. It was also found that with the increase in the PECH content, the cross-section of membranes 1–5 gradually became rough. This was due to the swelling caused by the presence of hydrophilic PECH quaternary ammonium groups in the membrane, which resulted in the rough cross-section of the membrane. Previous studies have shown that the cross-section roughness of membrane has a certain influence in acid recovery performance [38,39]. Membranes with a certain cross-section roughness have higher acid recovery performance [38,39]. In conclusion, the prepared membrane was suitable for the DD process of acid recovery, which was confirmed in subsequent DD experiments.

### 2.3. Thermal and Mechanical Properties of Membranes

The thermal stability of membranes 1–5 was tested by TGA. The test temperature was 20–800 °C in a N_2_ atmosphere. The TGA data from membranes 1–5 were collected and are presented in Figure 3. The weight loss of the membrane can be divided into three stages within the temperature range of 20 °C to 800 °C. The first is the weight loss at 20–200 °C, which was related to the evaporation of the membrane-bound water and solvent [40,41]. Membranes 1–5 had a certain weight loss, which showed the prepared membranes had a certain degree of hydrophilicity. The second is the degradation range of the quaternary ammonium groups, which was in the weight loss stage of 200–430 °C [36,40,41]. From Figure 3, it can be seen that the weight loss of membranes 4–5 was more than for membranes 1–3. Because the PECH content of membranes 4–5 was higher than for membranes 1–3, membranes 4–5 had more quaternary ammonium groups. The third is that the decomposition temperature of the polymer main chain was above 430 °C [41,42]. All of the membranes had good thermal stability below 200 °C. The good thermal stability of the membranes was due to their net structure. The above decomposition temperature was much higher than the operating temperature of the membranes, so the prepared membranes had good stability at operating temperature.

A diagram of the mechanical properties of membranes 1–5 is shown in Figure 4. The TS values of membranes 1–5 were 12.5–25.6 MPa, and the E_b_ values were 4.8–13.8%. From Figure 4, it can be observed that the mechanical properties of membranes 1–5 changed. The main reason for the change was the quaternary amination crosslinking reaction of BPPO and PECH with TMHD in the membrane. The mechanical properties and crosslinking degree of membranes 1–5 varied with the increase in the PECH components in the membrane. With the increase in the PECH content in the membrane, the hydrophilicity of membranes 1–5 increased. Thus, the quantity of water molecules in the membrane increased and played the role of plasticizers, and the E_b_ value of the membrane showed an increasing trend [36]. The decrease in TS was mainly due to the decreased compatibility of the membrane. The increase in the PECH content in the membrane led to an increase in the phase separation, which reduced the compatibility of the membrane and led to the decrease in the TS value of the membrane [43]. Compared with previously reported membranes (16.6–18.3 MPa [27]), the prepared membranes showed higher TS values. The prepared membranes should have high TS and E_b_ values. When comparing the TS and E_b_ values of membranes 1–5, membrane 3 had the best mechanical properties.

### 2.4. Acid Resistance

The acid resistance of membranes 1–5 was tested and the results are shown in Table 1. It was found that the weight of membranes 1–5 in acidic solution remained almost unchanged. The weight retention rate was 98.2–98.6% after 7 days and 98.1–98.8% after 14 days, with the weight retention rate remaining above 98%. PECH and BPPO had good chemical stability [36] and the net structure formed by PECH and BPPO crosslinking further enhanced the chemical stability of the membrane, which gave the membrane good acid resistance.

### 2.5. Water Uptake (W_R_) Linear Expansion Rate (LER) and Ion Exchange Capacity (IEC)

The W_R_, LER, and IEC data of membranes 1–5 were collected (Figure 5). The corresponding data from membranes 1–5 are shown in Figure 5a. The W_R_ values of the membranes were 30.05–91.04% and the W_R_ values of the membranes showed a growing trend. The LER values of the membranes were 9.8–25.4%.

At 25 °C, the water absorption of the membrane increased with the increase in the PECH content. BPPO is a hydrophobic polymer, so the water absorption of the membrane mainly depends on the quaternary ammonium groups [44,45]. The quaternary ammonium groups of PECH had high hydrophilicity, which greatly improved the hydrophilicity of the membranes, so the W_R_ value increased with the increase in PECH [44]. The LER values of membranes 1–5 increased with the increase in the PECH content in the membranes. Compared with other reported membranes, according to Table 2, the LER value was lower. Therefore, the prepared membranes had good dimension stability.

The IEC values of membranes 1–5 are shown in Figure 5b. The IEC values were 1.81–2.08 mmol/g. IEC indicated the density of the ion exchange groups in the membrane, which were used for the transport of ions. As the content of PECH increased, the quaternization sites of the membranes also gradually increased, so the number of ion exchange groups increased, which led to the increase in the IEC values. The higher IEC value of the membrane, the stronger the ion transport ability of the membrane [7,49]. Therefore, the U_H_^+^ and U_Fe_^2+^ values of the membranes also increased with the increase in IEC values (Figure 6). The IEC values confirmed the existence of ion exchange groups in the membranes.

High values of W_R_ and LER indicate the poor stability of a membrane. The higher the IEC value, the stronger the ion transport ability of the membrane. From comparing the W_R_, LER, and IEC values of membranes 1–5, membrane 3 had the best performance.

### 2.6. Diffusion Dialysis Performance

The diffusion dialysis performance of the membranes represented the application value [36]. The DD experiment results are presented in Figure 6. The U_H_^+^ of membranes 1–5 was 17.3–26.2 × 10^−3^ m/h at 25 °C. At 25 °C, the U_H_^+^ value increased with the increase in the PECH content. The U_Fe_^2+^ of membranes 1–5 was 0.7–0.97 × 10^−3^ m/h. The U_Fe_^2+^ value also increased with the increase in the PECH content. The gradual increase in the U_H_^+^ value was related to the increase in the ion group content in the membranes [25]. Membranes 1–5 contained ion exchange groups, which can exchange ions [49,50]. To maintain the electric charge balance, the cations also pass through the membrane. It is more difficult for Fe^2+^ ions to pass through membranes than H^+^ ions due to their larger volume and greater electric charge [36,50]. Therefore, the values of U_H_^+^ were greater than U_Fe_^2+^, and both showed an increasing trend at the same time [49,50].

The separation factor (S) is a ratio of U_H_^+^ to U_Fe_^2+^. The S values of membranes 1–5 are shown in Figure 6b. The range of S values was 24.6–27.0. Previous studies have shown that the separation factor is related to the morphology and functional groups of the membrane [49,51,52,53]. Therefore, the reasons for the increase in the membranes’ S value were as follows: firstly, with the increase in the PECH content of the membrane, the cross-section morphology of the membrane became uneven and rough, and according to Figure 2, the increase in the phase separation resulted in a decrease in the S value [54]. Secondly, as the PECH content increased in the membrane, the IEC value also increased gradually, which allowed the membrane to obtain a higher S value [53,54,55]. Thirdly, the increase in the degree of crosslinking with the increase in the PECH content led to the increase in the S value. Finally, due to the interaction between electric charges, the greater the IEC value of the membrane, the greater the mutual repulsion of Fe^2+^, which limited the transmission of Fe^2+^, so the S value of the membrane slightly increased [5,56,57,58]. Compared with DF-120B and previously reported membranes based on BPPO, as shown in Table 3, the prepared membranes had advantages in acid recovery by DD. When comparing the U and S values of the membranes, the S and U values of membranes 3–5 were greater than for membranes 1–2, which indicated membranes 3–5 had better performance.

## 3. Materials and Methods

### 3.1. Materials

Brominated poly (2,6-dimethyl-1,4-phenyleneoxide) (BPPO) was purchased from ShanDong TianWei Membrane Technology Co., Ltd. (Weifang, China). N,N,N,N-tetramethyl-1,6-hexanediamine (TMHD) was purchased from Sinopharm Chemical Reagent Co., Ltd. (Shanghai, China). Polyepichlorohydrin (PECH) was purchased from Sinopharm Chemical Reagent Co., Ltd. N-methylpyrrolidone (NMP) was purchased from Sinopharm Chemical Reagent Co., Ltd. Other materials used include potassium chromate (K_2_CrO_4_), sodium sulfate (Na_2_SO_4_), hydrochloric acid (HCl), trimethylamine (TMA), potassium permanganate (KMnO_4_), ferrous chloride (FeCl_2_.4H_2_O), and methyl orange (MO) from domestic chemical reagent companies. Deionized water was used throughout the experiment.

### 3.2. Reaction Equation Diagram for the Preparation of Ion Exchange Membranes

The AEM was prepared by the quaternization reaction of PECH and BPPO. The quaternization reagents were TMHD and TMA. The reaction equation diagram is shown in Figure 7.

### 3.3. Preparation Process of Anion Exchange Membranes

To begin with, BPPO was placed in 7.5 mL NMP and stirred at 60 °C for 2 h to ensure that the BPPO was completely dissolved. Then, PECH was added into the BPPO solution and stirred at 80 °C for 2 h to ensure that the BPPO and PECH were completely dissolved. Next, TMHD was added to the mixed solution at 80 °C and stirred for 1 h. The mixed solution was poured onto flat glass. The flat glass was placed into an oven at 80 °C and kept at this temperature for 24 h to make mixed solution happen fully thermal crosslinking reaction to form membrane. Then, the membrane was soaked in TMA for 24 h to improve the quaternization degree of membrane. After that, the membrane was rinsed 2–3 times with DI water. Finally, the membrane was dried at 60 °C for 24 h.

According to the above membrane preparation process, five membranes were prepared with the amount of experimental reagent listed in Table 4 and named membrane 1–5. According to the membrane preparation process, a membrane was prepared without adding any quaternary ammonium reagent, which was used for a contrast experiment and named BPPO/PECH film. The experimental process diagram is shown in Figure 8. The control experiment was designed to study the influence of PECH on membrane performance by changing the PECH content.

### 3.4. Membrane Test

#### 3.4.1. Membrane Morphology

The cross-sectional morphologies of AEM were determined using scanning electron microscopy (SEM) (JSM-7500F, JEOL, Tokyo, Japan).

#### 3.4.2. FTIR Analysis

The successful synthesis of AEM was demonstrated using an FTIR spectrometer (Vector 22, Bruker, Billerica, MA, USA) to obtain the attenuated total reflection (ATR) in the range of 4000–500 cm^−1^.

#### 3.4.3. Thermal and Mechanical Properties of Membranes

The thermal stability of AEM was determined using a Shimadzu TGA-50H analyzer with a constant heating ratio of 10 °C/min^−1^ from 20–800 °C in a N_2_ atmosphere.

The mechanical property of the membranes was collected using a microcomputer-controlled electronic testing machine (CMT4304, MTS Industrial Systems Co., Ltd., Eden Prairie, MN, USA). The speed of the machine crosshead was 25 mm/min.

#### 3.4.4. Acid Resistance

The acid resistance of the membranes was expressed as a function of the membrane weight loss after being kept in HCl for a period of time. The acid resistance test steps of the membranes were as follows: the first group of membranes were put in 2 mol/L HCl 7 days, and the second group of membranes were put in 2 mol/L HCl for 14 days. Then, the membrane was taken out of the HCl, washed with DI water, and dried. Finally, the weight data of the membrane were collected.

#### 3.4.5. Ion Exchange Capacity (IEC)

The experimental operation method was as follows: the membranes were immersed in 1.0 mol/L^−1^ NaCl solution for 24 h, during which all ions at ionic sites were changed into Cl^−^. Next, the membranes were rinsed using DI water to remove the NaCl residue. The washed membranes were immersed in 0.05 mol/L^−1^ Na_2_SO_4_ solution for 48 h. Finally, the amount of liberated Cl^−^ ions from the membranes was estimated by titration with 0.1 mol/L^−1^ AgNO_3_ and using K_2_CrO_4_ as an experiment indicator. The *IEC* of the membranes was calculated using this equation:(1)$$IEC=\frac{C\times V}{W_{dry}}$$
where *C* is the concentration of AgNO_3_ solution, *V* is the volume of AgNO_3_ solution, and *W_dry_* is the weight of the dry membrane.

#### 3.4.6. Water Uptake (*W_R_*) and Linear Expansion Rate (LER)

The membrane was soaked in DI water and kept for 24 h to ensure it had fully absorbed the water. Then, the residual water was removed from the membrane surface, and the membrane was weighted again to collect the wet weight of the membrane (*W_wet_*). After that, the membrane was dried at 60 °C for 24 h, and weighed to obtain the dry weight of membrane (*W_dry_*). The water absorption (*W_R_*) of the membrane was calculated using the following equation:(2)$$W_{R}=\frac{W_{wet}-W_{dry}}{W_{dry}}\times 100\%$$
where *W_wet_* is the weight of the wet membrane and *W_dry_* is the weight of the dry membrane.

Membranes 1–5 were made into samples, which were 2 cm long and 1 cm wide, for the experiment. The membrane samples were immersed in water at 25 °C for 24 h, and then removed to collect the corresponding data. The *LER* value of the membranes were calculated according to the corresponding calculation formula. The calculation formula of *LER* was the following equation:(3)$$LER=\frac{\left(L_{wet-}L_{dry}\right)}{L_{dry}}\times 100\%$$
where *L_wet_* is the length of the wet membrane and *L_dry_* is the length of the dry membrane.

#### 3.4.7. Diffusion Dialysis Performance (DD)

The diffusion dialysis test was carried out using a custom device consisting of two chambers: one chamber was the feed side and the other chamber was the permeate side. Before testing, the membrane was kept in a solution of HCl (1.0 mol/L^−1^)/FeCl_2_ (0.2 mol/L^−1^) at 25 °C for 12 h. Then, the membrane was rinsed with DI water. After that, the membrane was placed between the two chambers. The feed side chamber was filled with 100 mL HCl (1.0 mol/L^−1^)/FeCl_2_ (0.2 mol/L^−1^) solution and the permeate side chamber was filled with 100 mL DI water. The experiment was carried out by stirring the two chambers at 25 °C for 45 min. After that, H^+^ was determined by titration with Na_2_CO_3_ (0.05 mol/L^−1^) solution at 25 °C. Potassium permanganate (0.002 mol/L^−1^) solution was used as a titrant for Fe^2+^ detection. Finally, the experimental data of the membrane were collected to calculate the results according to the formula. The experimental device and schematic diagram are shown in Figure 9.

The dialysis coefficient (*U*) of a single component was calculated as follows:(4)$$U=\frac{M}{A_{t}∆C}$$
where *M* is the number of moles with one single component transferred from the feed side to the permeate side, *A* is the effective area of the measured membrane, and *t* is the experiment time.

Δ*C* is the logarithmic average concentration of the two sides calculated according to the following equation:(5)$$∆C=\frac{C_{f}^{0}-\left(C_{f}^{t}-C_{d}^{t}\right)}{\mathrm{Ln}\left[C_{f}^{0}-\left(C_{f}^{t}-C_{d}^{t}\right)\right]}$$
where $C_{f}^{0}$ is the solution concentration on the feed side at time 0 and $C_{f}^{t}$ is the solution concentration on the feed side at time *t*. $C_{d}^{t}$ is the solution concentration on the osmotic side at time *t*.

The separation coefficient (*S*) between acid and salt is calculated as follows:(6)$$S=\frac{U_{H}}{U_{Fe}}$$

## 4. Conclusions

We prepared a series of net structure anion exchange membranes based on BPPO/PECH. The network structure generated by the quaternization reaction of TMHD with BPPO and PECH improved the mechanical properties and dimensional stability of the membranes. The prepared AEM showed excellent mechanical properties (TS: 12.5–25.6 MPa, E_b_: 4.8–13.8%), thermal performance, and acid stability (weight retention rate: 98.2–98.6%), as well as a good linear swelling ratio (LER: 9.8–25.4%) and hydrophilicity (W_R_: 30.05–91.04%). The acid dialysis coefficients (U_H_^+^) of membranes 1–5 were 0.0173–0.0262 m/h at 25 °C. The separation factors (S) of membranes 1–5 were 24.6–27.0 at 25 °C. By comparing the relevant experimental data of membranes 1–5, membrane 3 had the best comprehensive performance. The acid recovery rate of the membrane prepared in this study was better than DF-120B (0.004 m/h) and other reported BPPO membranes. In summary, we prepared a new anion exchange membrane that had good application potential in acid recovery via DD.

## Figures and Tables

**Figure 1 ijms-24-08596-f001:**
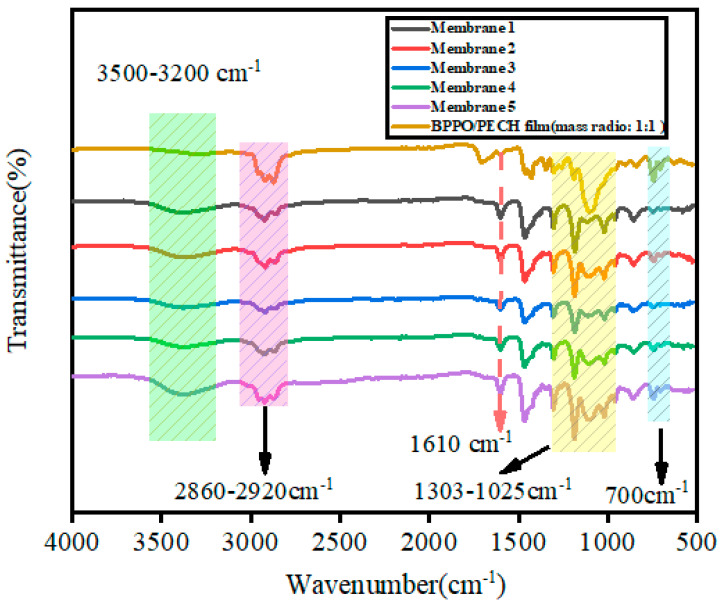
FTIR spectra of membranes 1–5 and BPPO/PECH film.

**Figure 2 ijms-24-08596-f002:**
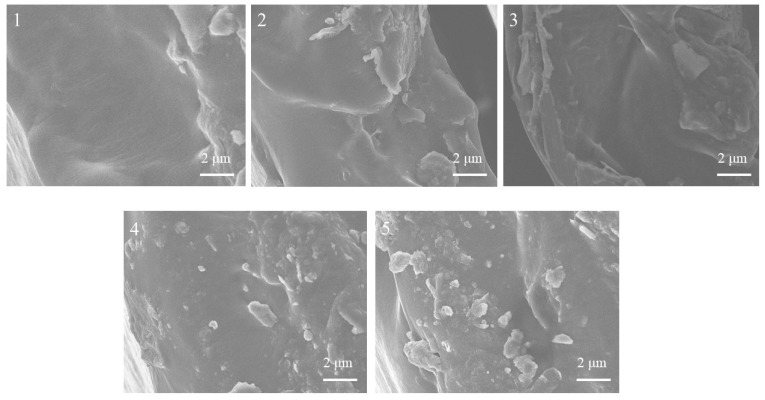
Cross-sections SEM images of membranes 1–5.

**Figure 3 ijms-24-08596-f003:**
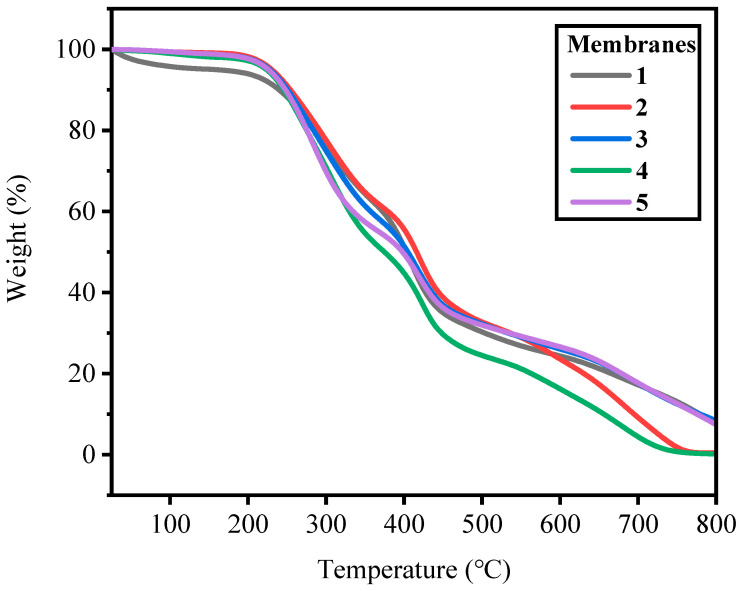
TGA diagrams of membranes 1–5.

**Figure 4 ijms-24-08596-f004:**
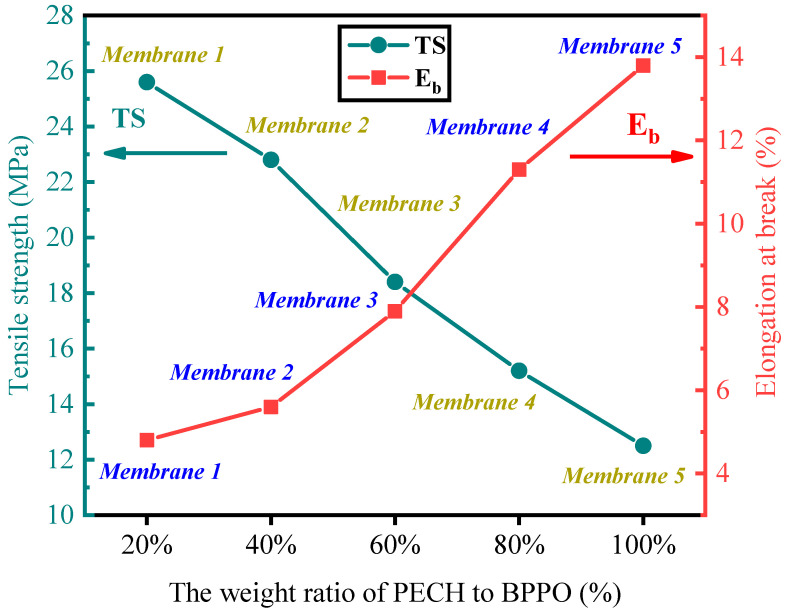
Mechanical properties of membranes 1–5.

**Figure 5 ijms-24-08596-f005:**
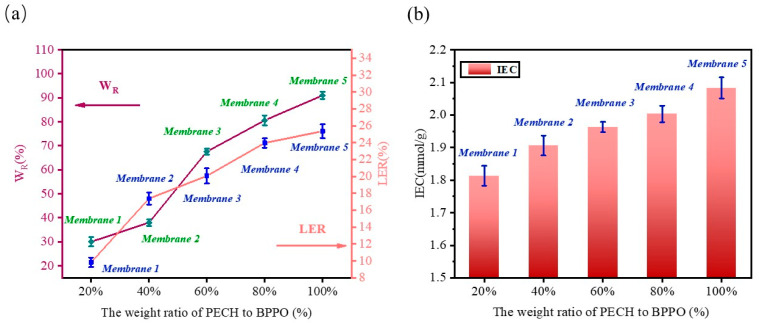
(**a**) Water uptake and linear expansion ratio of membranes 1–5, (**b**) ion exchange capacity of membranes 1–5.

**Figure 6 ijms-24-08596-f006:**
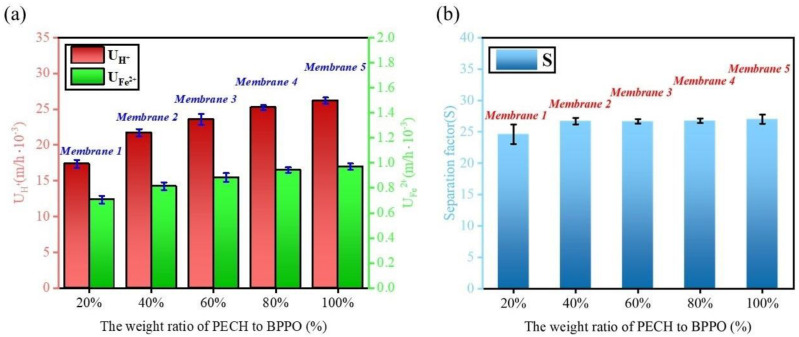
(**a**) Acid dialysis coefficient (U_H_^+^) and permeability coefficient of iron ions of membranes 1–5, (**b**) separation coefficient (S) of membranes 1–5.

**Figure 7 ijms-24-08596-f007:**
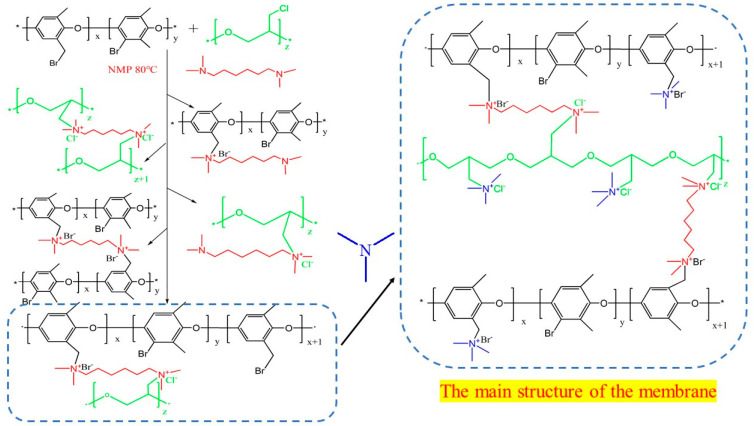
Reaction equation diagram for the preparation of ion exchange membranes.

**Figure 8 ijms-24-08596-f008:**
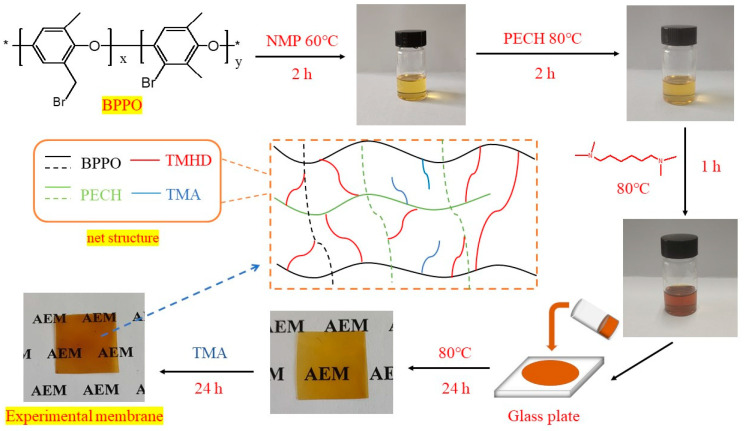
Experimental process diagram.

**Figure 9 ijms-24-08596-f009:**
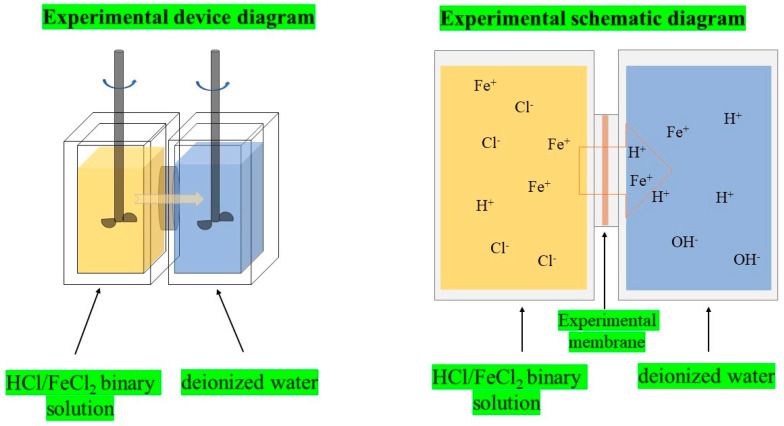
Experimental device and schematic diagram.

**Table 1 ijms-24-08596-t001:** Acid resistance data diagram of membranes.

Weight Retention Rate/%	Membrane 1	Membrane 2	Membrane 3	Membrane 4	Membrane 5
7 d	98.6	98.5	98.4	98.5	98.2
14 d	98.8	98.6	98.4	98.5	98.1

**Table 2 ijms-24-08596-t002:** Comparison of WR, LER, and IEC values of different membranes.

Membranes	W_R_/%	LER/%	IEC/mmol·g^−1^	Refs.
Membranes 1–5	30.05–91.04	9.8–25.4	1.81–2.08	This work
PVA/EPTAC	196–267	35–44	0.58–1.15	[46]
QDAB/PVA	47.8–71.3	68.2–204.6	0.86–1.46	[47]
PVA/THOPS	38–86	15–25	0.70–1.56	[48]

**Table 3 ijms-24-08596-t003:** U_H_^+^ and S of membranes in comparison to other reported membranes.

Membrane	U_H_^+^ (10^−3^ m/h)	S	Refs.
Membranes 1–5	17.3–26.2	24.6–27.0	This work
BPPO/DMAP	0.37–20.3	73–351	[25]
BPPO/TPA	5.6–10.4	21.9–38.8	[35]
BPPO/TMA	4.3–12	13.14–32.87	[53]
BPPO/PEI	63–70	11–20	[56]
DF-120B	4.00	24.30	[27]

**Table 4 ijms-24-08596-t004:** Composition of membranes 1–5 and BPPO/PECH film.

Number	Membrane 1	Membrane 2	Membrane 3	Membrane 4	Membrane 5	BPPO/PECH Film
BPPO	0.25 g	0.25 g	0.25 g	0.25 g	0.25 g	0.25 g
PECH	0.05 g	0.10 g	0.15 g	0.20 g	0.25 g	0.25 g
TMHD	0.12 g	0.14 g	0.16 g	0.18 g	0.20 g	
TMA	0.02 g	0.02 g	0.02 g	0.02 g	0.02 g	

## Data Availability

The study did not report any data.

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
