# Peer review of "Anion Exchange Membrane Based on BPPO/PECH with Net Structure for Acid Recovery via Diffusion Dialysis"

_ijms, 2023, doi:10.3390/ijms24108596_

Round 1

Reviewer 1 Report

The manuscript contains new experimental results, so it can be published.

Before that, the manuscript should be carefully checked. Here are some comments.

„…DD process is more environmental friendly and efficiently in wastewater recovery” – comparing to what?

 „The application effect of the prepared membrane in the DD process are tested also.” – what is „application effect”? are -> is.

 „2.2. The reaction equation of ion exchange membrane” - ?

 2.4.3. Thermodynamic properties of membranes 131

„Thermal stability of the membranes …

The mechanical property …”

Are these properties thermodynamic? Maybe “Thermal and mechanical properties”?

 „firstly, the prepared anion exchange membrane was put into 2 mol/L HCl aqueous solution at 25 ℃. Then, after immersing in the solution for 7 day or 14 day, the membrane was taken out, washed and dried” – „the solution for 7 day” was different from 2 M HCl or the same? If the same then it would be better to write shortly: „After immersing in 2 M HCl solution for 7 days …”

days, not day.

 „two chambers were filled with 100 mL HCl/FeCl2 solution” – what was concentration of HCl and of FeCl2?

“HCl/FeCl2 solution concentration was determined by titration with standard Na2CO3 solution” – what does it mean “standard”? How Fe2+ was determined?

 “the concentration of HCl or FeCl2 on the osmotic side” – what does it mean “osmotic side”?

“certain phase separation occurred in the membrane.” – what are the phases?

 “and increased the recovery efficiency of H+ ions significantly[36,37]. At same the time, Fe2+ ions also increased[36]” – what increased?

 Figs. 8, 9 – what ratio? weight? molar? Looking at Fig.1 the best would be the ratio of Cl (PECH) to Br (BPPO).

 “and stronger electric charge” – stronger charge?? greater.

“The separation factors (S) were 24.6-27.0 “ – there is one separation factor and many of its values.

“the reported acid recovery value of the membrane” - “the reported acid recovery of the membrane”.

 “the value of UH+ was greater than the value of UFe2+,” squeeze to: “UH+ was greater than UFe2+,”

 Conclusions

“The membrane has tremendous application potential in acid recovery via DD. “ – I wouldn’t be so optimistic. In Tab. 4, there are membranes of much higher selectivity, e.g. Porous BPPO S = 81-665 which is 3-20 times greater than S of BPPO/PECH. U(H+) of these two membranes are comparable, thus the dialysis process with Porous BPPO is equally fast but much more selective than using BPPO/PECH.

 English should be improved, e.g.:

 „BPPO and PECH were occur quaternization reaction” - ?

„The prepared membrane on the flat glass was put into oven, and keep temperature at 80 ℃ for 24 hours to make fully thermal crosslinking reaction of the membrane.”

„According to the collected data, calculate the average value of the measurement data to reduce the experimental error.”

„was measured by the Mohr method” – determined. And the Mohr method refers only to the Cl- titration, not to the determination of IEC.

„Calculate the average value of the collected data according to the above formula to reduce experimental error.”

“has showed”.

These are just a few examples.

Author Response

Reviewer 1

  1. The comparison with BPPO based commercial membranes hinted in the abstract is not given in the results, in order to establish the challenges, overcome (or not) by the novel developments carried about by the authors.

The main problem associated with AEM is its hydrophily, which limits the Ion transport rate in the DD process. The author has added a comparison of other membranes and the challenges to be overcome in the discussion and results. The author has corrected it. Thank you for your correction.

2.Line 12, page 1, what do the authors mean by “machine performance”?

This refers to the performance of the membrane's tensile strength, elongation at break, and linear expansion ratio. Thank you for your question.

3.Line 40, page 1, the acronym for cation exchange membrane is CEM not CAM, what do the authors mean?

This is the author's writing error, which has been corrected. Thank you for your correction.

  1. Line 53, page 2, what do the authors mean by “excellent performance”?

This refers to the excellent performance of the membrane in terms of heat resistance, acid resistance, mechanical properties, ion transport performance, and other performance. Thank you for your question.

  1. Line 56, page 2, what do the authors mean by “raw materials” there are anion exchange membranes prepared from different types of materials (inorganic, polymeric) although those based on different polymers are the most common. Please be more specific.

This refers to the main chain material of the polymer used to prepare anion exchange membranes, which has been corrected. Thank you for your correction.

  1. 6.Line 77, “were occur”? please correct.

This refers to the quaternary ammonium chemical reaction between BPPO and PECH. The author has corrected it. Thank you for your correction.

  1. 7.Lines 76-84, page 2, The techniques of IR, IEC, WU, swelling and morphology selected for characterization are adequately selected but the description should be improved in order to evaluate the results obtained there from. For instance, sample sizes in TGA (line 13, page 4), or the selection of atmosphere for the thermogravimetric analyses…

The author has improved the description of the test. Thank you for your correction.

  1. 8.Page 3, table 1, why the base membranes does not contain TMHD or TMA as the blended membranes? A control experiment would add insight as to what components are really influencing the performance.

The BPPO/PECH film was used as the control group for infrared reaction. The quaternized infrared characteristic peaks were compared by not adding TMHD and TMA to the base membrane. Thank you for your question.

The control experiment is to study the effect of PECH on the performance of the membrane by changing the PECH content. The author has corrected it. Thank you for your correction.

  1. 9.Page 5, line 18, is 60ºC a sufficient temperature to dry all the water content in the membrane for the water uptake analyses? Please justify.

The water and most of the bound water of the membrane were removed from the surface by drying in a 60 ℃ oven for a sufficient period of time. Residual water in the membrane is difficult to remove, which has minimal impact. The value of membrane can be calculated average value through multiple experiments to reduce its impact on the data. Thank you for your question.

10.Page 6, line 19, why do the authors use the mean logarithmic concentration here? Please justify.

It can reduce the impact of errors during the experimental process. Avoid the influence of concentration polarization on the experiment. Thank you for your question.

  1. 1Page 7, line 28, “peak of C-N exist represent”. Please revise English expression throughout the text.

The author's expression is incorrect and has been corrected. Thank you for your correction.

  1. 1Page 8, figure 5 and text below. What position across the membrane thickness were the SEM images taken from? Are there differences near the bottom, near the surface or in the middle that can provide information on the homogeneity? Please clarify. What is the whole thickness of the membrane samples?

SEM images are images of the fracture surface of the membrane. The image shooting position is in the middle of the fracture surface of the membrane, and there is no significant difference in images from different positions. It can be seen from the figure that the membrane is uneven. The thickness of the membrane is 300 microns. Thank you for your question.

  1. 1Page 9, figure 6 and text below. The differences between the curves are very slight, please try to quantify some information on the weight losses and key onset temperatures mentioned slightly in the text to extract some information between membrane 1 to 5 synthesized in this work.

The author has added corrections in the article. Firstly, the weight loss around at 20-200 ℃, which was related to the evaporation of the membrane bound water and solvent. Membranes 1-5 have a certain weight loss, indicating that the prepared membranes have a certain degree of hydrophilicity. Secondly, the degradation range of quaternary ammonium groups was in the weight loss stage of 200-430 ℃. From the figure, it can be seen that the weight loss of membrane 4-5 was more than that of membrane 1-3. This is because the PECH content of membrane 4-5 is higher than that of membrane 1-3, so membrane 4-5 has more quaternary amination groups. Thirdly, the decomposition temperature of the polymer backbones was above 430 ℃. Thank you for your correction.

  1. 1Page 10, line 26, define Eb. Compare with other BPPO membranes in literature and commercially available membranes as the one mentioned in the abstract of the present paper.

When an object is pulled apart by an external force, the ratio of the elongation after stretching to the length before stretching is called the elongation at break(Eb). The author has added a comparison between the prepared membrane and other BPPO membranes. The author has corrected it. Thank you for your correction.

  1. 1Page 11,line 29, “the LER value of membrane was also grew”, how is this observed in Table 3? Please clarify and correct.

This refers the LER value of the prepared membrane 1-5 shows an increasing trend. The author has corrected it. Thank you for your question.

  1. 1Tables 3 and 4. The membranes 1 to 5 are grouped together in a single line, not allowing to extract information on the influence of composition on the characteristics measured by the authors. Please revise.

The author has corrected it. Thank you for your correction.

  1. 1Page 13, line 35, ho w is the PECH crosslinking affecting, which is the best membrane composition and the conclusions of the study?

The author has corrected it. Thank you for your correction.

Through comparison of relevant experimental data, membrane 3 has the best comprehensive performance. The author has corrected it. Thank you for your correction.

This proves that the effect of PECH on the enhancement of membrane performance is effective. The results show that the prepared membrane has good application potential in acid recovery by DD. The author has corrected it. Thank you for your correction.

Reviewer 2 Report

This manuscript deals with the synthesis of modified BPPO membranes and their characterization for use in acid recovery by diffusion dialysis. Since most of the anion exchange membranes reported so far and some of those commercially available are based on BPPO the starting point of this work is worthy of studying to improve membrane and process performance.

However, it is not clear how the membranes prepared by the authors perform in the present applications since there are several serious misconceptions between anion exchange and cation exchange membranes, and the membranes are characterized by the proton conductivity (page 2, line 63, line 72) when they are supposed to be anion-exchange membranes. The comparison with BPPO based commercial membranes hinted in the abstract is not given in the results, in order to establish the challenges, overcome (or not) by the novel developments carried about by the authors. 

Other comments:

Line 12, page 1, what do the authors mean by “machine performance”?

Line 40, page 1, the acronym for cation exchange membrane is CEM not CAM, what do the authors mean?

Line 53, page 2, what do the authors mean by “excellent performance”?

Line 56, page 2, what do the authors mean by “raw materials” there are anion exchange membranes prepared from different types of materials (inorganic, polymeric) although those based on different polymers are the most common. Please be more specific.

Line 77, “were occur”? please correct.

Lines 76-84, page 2, The techniques of IR, IEC, WU, swelling and morphology selected for characterization are adequately selected but the description should be improved in order to evaluate the results obtained there from. For instance, sample sizes in TGA (line 13, page 4), or the selection of atmosphere for the thermogravimetric analyses…

Page 3, table 1, why the base membranes does not contain TMHD or TMA as the blended membranes? A control experiment would add insight as to what components are really influencing the performance.

Page 5, line 18, is 60ºC a sufficient temperature to dry all the water content in the membrane for the water uptake analyses? Please justify.

Page 6, line 19, why do the authors use the mean logarithmic concentration here? Please justify.

Page 7, line 28, “peak of C-N exist represent”. Please revise English expression throughout the text.

Page 8, figure 5 and text below. What position across the membrane thickness were the SEM images taken from? Are there differences near the bottom, near the surface or in the middle that can provide information on the homogeneity? Please clarify. What is the whole thickness of the membrane samples?

Page 9, figure 6 and text below. The differences between the curves are very slight, please try to quantify some information on the weight losses and key onset temperatures mentioned slightly in the text to extract some information between membrane 1 to 5 synthesized in this work.

Page 10, line 26, define Eb. Compare with other BPPO membranes in literature and commercially available membranes as the one mentioned in the abstract of the present paper.

Page 11,  line 29, “the LER value of membrane was also grew”, how is this observed in Table 3? Please clarify and correct.

Tables 3 and 4. The membranes 1 to 5 are grouped together in a single line, not allowing to extract information on the influence of composition on the characteristics measured by the authors. Please revise.

Page 13, line 35, ho w is the PECH crosslinking affecting, which is the best membrane composition and the conclusions of the study?

Author Response

Reviewer 2

  1. 1.There are lot of grammatical mistakes in the paper. The language needs to corrected throughly.

The author has looked for and corrected related syntax error. Thank you for your correction.

  1. 2.The explanation of SEM cross-section of the membranes are completely wrong.

The author has further improved and discussed this part of the article. Thank you for your correction.

  1. 3.I suggest to delete the blue dotted arrows from figure 7.

The author has modified the graphics. Thank you for your correction.

  1. 4.The authors did not make any effort to correlate the observed properties presented in figure 8with those of figure 9. I strongly suggest to improve the discussion in this regard.

The author has further improved and discussed this part of the article. Thank you for your correction.

Reviewer 3 Report

The membrane material design concept presented in the paper is rather old. The experimental approach is also quiet regular. As a result the overall scietific outcome of the paper is not really something exceptional. In terms of novelty the article is has rather low ranking. Considering the detail nature of the experimental work I suggest to publish the paper. Mostly I have major corrections- 

1. There are lot of grammatical mistakes in the paper. The language needs to corrected throughly. 

2. The explanation of SEM cross-section of the membranes are completely wrong. I the explanation of Figure 5 the authors stated - ", the cross-section morphologies of the membrane 1- 5 also showed that BPPO and PECH had certain compatibility. It can be seen from Figure 5 that with the increase of PECH content, the compatibility of the membrane decrease, and the cross-section of the membrane becomes uneven gradually, a certain phase separation occurred in the membrane. The slight phase separation increased the ion channel of membrane"

Figure 5 presents secondary electron images of fractured film. None of the above mentioned statements can be made from these images. The observed uneveness in the cross-section simply comes from sample preparation. In my opinion these images doesnot say anything about the compatibility oft BPPO and PECH and phase separation of the membrane. Such wrong explanation must be deleted before publication.

3. I suggest to delete the blue dotted arrows from figure 7. 

4.  The authors did not make any effort to correlate the observed properties presented in figure 8  with those of figure 9. I strongly suggest to improve the discussion in this regard.

Author Response

Reviewer 3

1.DD process is more environmental friendly and efficiently in wastewater recovery” – comparing to what?

By compare with other membrane separation processes. Thank you for your question.

2.The application effect of the prepared membrane in the DD process is tested also.” – what is „application effect’’?

This refers to the performance testing of the prepared membrane in the DD method for acid recovery, such as acid dialysis coefficient, separation coefficient, etc. Thank you for your question.

3.The reaction equation of ion exchange membrane” - ?

This refers to the reaction equation diagram for the preparation of ion exchange membranes. The author has corrected it. Thank you for your correction.

4.Thermodynamic properties of membranes’’?

This refers to the thermal and mechanical properties of the membrane. The author has corrected it. The author has corrected it. Thank you for your correction.

  1. 5.firstly, the prepared anion exchange membrane was put into 2 mol/L HCl aqueous solution at 25 ℃. Then, after immersing in the solution for 7 day or 14 day, the membrane was taken out, washed and dried” – „the solution for 7 day” was different from 2 M HCl or the same? If the same then it would be better to write shortly: „After immersing in 2 M HCl solution for 7 days …”days, not day.

This refers to soaking in 2 M HCl aqueous solution for 7 or 14 days. Thank you for your correction and the author has corrected it.

  1. two chambers were filled with 100 mL HCl/FeCl2solution” – what was concentration of HCl and of FeCl2?

The concentration of HCl is 1.0 mol/L-1, and the concentration of FeCl2 is 0.2 mol/L-1. The author has corrected it. Thank you for your correction 

  1. 7.“HCl/FeCl2solution concentration was determined by titration with standard Na2CO3 solution” – what does it mean “standard”? How Fe2+ was determined?

This standard refers to the concentration of Na2CO3 solution is 0.05 mol/L-1. 0.002mol/L-1 potassium permanganate solution as a titrant for Fe2+detection. The author has corrected it. Thank you for your correction 

  1. 8.the concentration of HCl or FeCl2on the osmotic side” – what does it mean “osmotic side”?

This refers to the side of the chamber containing deionized water in the diffusion dialysis experimental device. Thank you for your question.

  1. 9.“certain phase separation occurred in the membrane.” – what are the phases?

According to the different forms and distributions of substances present in the system, the system is divided into phases. Phase refers to the aggregation state of uniform substances with identical physical and chemical properties and components, without external force. Thank you for your question.

  1. 1“and increased the recovery efficiency of H+ions significantly[36,37]. At same the time, Fe2+ ions also increased[36]” – what increased?

This refers to an increase in the rate of Fe2+ions passing through the membrane. Thank you for your correction and the author has corrected it.

11.Figs. 8, 9 – what ratio? weight? molar? Looking at Fig.1 the best would be the ratio of Cl (PECH) to Br (BPPO).

This refers to the weight ratio of PECH and BPPO. Thank you for your correction and the author has corrected it.

  1. 1“and stronger electric charge” – stronger charge?? greater.

The author has corrected it. Thank you for your correction.

  1. 1“The separation factors (S) were 24.6-27.0 “ – there is one separation factor and many of its values.

This refers to the separation factors of membranes 1-5 were 24.6-27.0. The author has corrected it. Thank you for your correction.

  1. 1“the reported acid recovery value of the membrane” - “the reported acid recovery of the membrane”.

The author has corrected it. Thank you for your correction.

  1. 1“the value of UH+was greater than the value of UFe2+,” squeeze to: “UH+ was greater than UFe2+,”

The author has corrected it. Thank you for your correction.

16.“The membrane has tremendous application potential in acid recovery via DD. “?

Compared to porous membranes, the prepared membranes are more stable and have a longer service life. Similarly, because the membrane does not have many pores, its transmission performance is slightly lower than that of porous membranes. Thank you for your question.

  1. 1Syntax error

The author has added corrections in the article. Thank you for your correction.

  1. 1„was measured by the Mohr method” – determined. And the Mohr method refers only to the Cl- titration, not to the determination of IEC.

The author has added corrections in the article. Thank you for your correction.

Round 2

Reviewer 2 Report

The authors have attended to most of reviewers' coments and the manuscript has been improved therefrom. Language editing might need thorough revision still, because there are several incorrections throughout the text, leading to several imprecisions as "Compared with OTHER membrane separation processes, DD process has certain advantages in recovering acid from industrial wastewater", etc.

Regarding the results, Table 4 distinguishes membrane 1 to 5 in composition of the blends, but Tables 2 and 3 collects the properties together so it is difficult for the reader to determine by himself the same conclusion stated by the authors in the answer to previous comments and the conclusions. Please correct.

Author Response

Review 2

  1. 1.Language editing might need thorough revision.

The author has made corrections to the language description of the article. Thank you for your correction.

  1. The interpretation of SEM figure can be further improved.

The roughness of membrane was due to the swelling caused by the presence of hydrophilic PECH quaternary ammonium groups in the membrane. Previous studies had shown that the cross-section roughness of membrane had a certain influence in acid recovery performance. The membrane with a certain cross-section roughness had higher acid recovery performance. The author has made corrections to the language description of the article. Thank you for your correction.

  1. Regarding the results, Table 4 distinguishes membrane 1 to 5 in composition of the blends, but Tables 2 and 3 collects the properties together so it is difficult for the reader to determine by himself the same conclusion stated by the authors in the answer to previous comments and the conclusions.

The prepared membrane should have high tensile strength and elongation at break. By comparing the TS and Eb values of membranes 1-5, membrane 3 had the best mechanical properties.

The high values of WR and LER indicated poor stability of the membrane. The higher the IEC value, the stronger the ion transport ability of the membrane. By comparing the WR, LER, and IEC values of membranes 1-5, Membrane 3 had the best performance.

By comparing the U and S values of membrane 1-5, the S and U values of membrane 3-5 were greater than membrane 1-2, which indicated the membrane 3-5 had better performance.

The author has added comparative analysis of membranes 1-5 in the corresponding detection experimental analysis, further improving the statement of the conclusion through these comparative analyses. Thank you for your correction.

Round 3

Reviewer 2 Report

Please clarify the aim of the paper, if it is the synthesis and preparation/modification of polymer membranes, and you have prepared5 different membranes and characterise them, why the results are not shown in the paper. Tables 2 and 3 report an average of the 5 membranes prepared by the authors and compare this average with literature, then conclusions are based on observations that are not shown in the manuscript. Please reconsider before submission.

Author Response

Review 2

  1. Please clarify the aim of the paper, if it is the synthesis and preparation/modification of polymer membranes, and you have prepared5 different membranes and characterise them, why the results are not shown in the paper. There are some related syntax error.

By adding PECH to improve the performance of BPPO membrane, AEM for DD was prepared. I have corrected the related syntax error in the article. Thank you for your correction.
